# Decoding the Role of Interface Engineering in Energy Transfer: Pathways to Enhanced Efficiency and Stability in Quasi-2D Perovskite Light-Emitting Diodes

**DOI:** 10.3390/nano15080592

**Published:** 2025-04-12

**Authors:** Peichao Zhu, Fang Yuan, Fawad Ali, Shuaiqi He, Songting Zhang, Puyang Wu, Qianhao Ma, Zhaoxin Wu

**Affiliations:** 1Key Laboratory for Physical Electronics and Devices of the Ministry of Education & Shaanxi Key Lab of Information Photonic Technique, School of Electronic Science and Engineering, Xi’an Jiaotong University, Xi’an 710049, China; zpc360360@stu.xjtu.edu.cn (P.Z.); fawad_ali@stu.xjtu.edu.cn (F.A.); shuaiqihe@stu.xjtu.edu.cn (S.H.); zhangsongting@stu.xjtu.edu.cn (S.Z.); 1933574811@stu.xjtu.edu.cn (P.W.); maqh2018@stu.xjtu.edu.cn (Q.M.); 2Collaborative Innovation Center of Extreme Optics, Shanxi University, Taiyuan 030006, China

**Keywords:** light-emitting diodes, quasi-2D perovskites, interface engineering, energy transfer

## Abstract

Quasi-two-dimensional (quasi-2D) perovskites have emerged as a transformative platform for high-efficiency perovskite light-emitting diodes (PeLEDs), benefiting from their tunable quantum confinement, high photoluminescence quantum yields (PLQYs), and self-assembled energy funneling mechanisms. This review systematically explores interfacial energy transfer engineering strategies that underpin advancements in device performance. By tailoring phase composition distributions, passivating defects via additive engineering, and optimizing charge transport layers, researchers have achieved external quantum efficiencies (EQEs) exceeding 20% in green and red PeLEDs. However, challenges persist in blue emission stability, efficiency roll-off at high currents, and long-term operational durability driven by spectral redshift, Auger recombination, and interfacial ion migration. Emerging solutions include dual-cation/halogen alloying for bandgap control, microcavity photon management, and insulator–perovskite–insulator (IPI) architectures to suppress leakage currents. Future progress hinges on interdisciplinary efforts in multifunctional material design, scalable fabrication, and mechanistic studies of carrier–photon interactions. Through these innovations, quasi-2D PeLEDs hold promise for next-generation displays and solid-state lighting, offering a cost-effective and efficient alternative to conventional technologies.

## 1. Introduction

Metal halide perovskites have recently emerged as a revolutionary class of optoelectronic materials, attracting unprecedented attention for their outstanding performance in solar cells, light-emitting diodes (LEDs), and photodetectors [1,2,3,4,5]. Among these, quasi-two-dimensional (quasi-2D) perovskites, with their layered quantum-well structures, have rapidly gained prominence in electroluminescent devices due to their tunable emission spectra, high photoluminescence quantum yields (PLQYs), and solution processability [6,7,8]. These materials ingeniously combine the advantages of low-dimensional quantum confinement effects with the structural stability of three-dimensional (3D) perovskites, offering a versatile platform for next-generation light-emitting technologies.

Although traditional 3D perovskites exhibit excellent charge transport properties, they suffer from intrinsic limitations, such as low radiative recombination efficiency, ion migration-induced instability, and spectral shifting under operational conditions [9,10,11]. In contrast, quasi-2D perovskites mitigate these challenges through their self-assembled, multi-quantum-well architectures. By incorporating bulky organic cations (e.g., phenethylammonium or butylammonium) into the perovskite lattice, these materials form natural energy funneling pathways, where excitons or free carriers rapidly transfer from wide-bandgap domains (low *n*-value phases) to narrow-bandgap regions (high *n*-value phases) [12,13,14]. This unique energy transfer mechanism not only suppresses nonradiative recombination but also enables precise spectral tuning across the visible to near-infrared spectrum, making them ideal candidates for high-purity color displays and energy-efficient lighting [15,16].

The competitive landscape of emissive materials further underscores the significance of quasi-2D perovskites. Organic light-emitting diodes (OLEDs), despite their commercial success in displays, face challenges such as complex vacuum-based fabrication, high costs, and efficiency roll-off at high brightness [17,18]. Quantum dot LEDs (QLEDs), while promising in color saturation, grapple with toxicity concerns, intricate synthesis, and environmental instability [19,20]. Quasi-2D perovskite LEDs (PeLEDs), on the other hand, leverage low-temperature solution processing, earth-abundant constituents, and exceptional defect tolerance, positioning them as cost-effective and scalable alternatives. Recent breakthroughs have demonstrated external quantum efficiencies (EQEs) exceeding 20% in green and red PeLEDs, rivaling state-of-the-art OLEDs and QLEDs [21,22,23,24,25,26].

However, the full potential of quasi-2D PeLEDs remains untapped, particularly in achieving stable blue emission, mitigating efficiency roll-off, and ensuring long-term operational stability [27,28]. A critical yet underexplored frontier lies in the rational engineering of interfacial energy transfer processes. Interfaces in PeLEDs, including perovskite/charge transport layer heterojunctions and intramaterial grain boundaries, are crucial for carrier injection, exciton dissociation, and photon out-coupling [29,30,31]. Improper interface management can result in uneven charge transport, increased interfacial recombination losses, and accelerated device degradation. As a result, cutting-edge methods like phase distribution control, additive engineering, and multi-functional interface modification have emerged as indispensable tools for optimizing energy transfer efficiency and device performance [32,33,34].

This review systematically examines the latest advancements in interfacial energy transfer regulation within quasi-2D PeLEDs, with a focus on material design, device architecture, and new insights into how things work. We elucidate the fundamental properties of quasi-2D perovskites, emphasizing their energy funneling dynamics and carrier recombination kinetics. Subsequently, we look at state-of-the-art methods for improving the efficiency of energy transfer at both the material and device levels. These include defect passivation, composition engineering, and interfacial energy alignment. Finally, we address persistent challenges and outline future directions for bridging the gap between laboratory-scale innovation and industrial-scale applications. Using ideas from different fields, such as materials science, photophysical mechanisms, and device engineering, this work aims to provide a comprehensive roadmap for advancing quasi-2D PeLEDs toward their ultimate realization in high-performance optoelectronic systems.

## 2. Fundamental Properties of Quasi-2D Perovskite Materials

Quasi-2D perovskites exhibit unique structural and photophysical properties that are pivotal for optoelectronic applications. Incorporating large organic cations (e.g., phenethylammonium or butylammonium) into the 3D perovskite lattice enables the spontaneous formation of layered quantum-well structures. The general chemical formula for quasi-2D perovskites is defined as A′_2_A_n−1_B_n_X_3n+1_ (1 ≤ *n* ≤ ∞, where A′ represents the large organic spacer cation, A is a small monovalent cation (e.g., methylammonium (MA^+^), formamidinium (FA^+^), or cesium (Cs^+^)), B denotes a divalent metal ion (e.g., Pb^2+^ or Sn^2+^), and X is a halide anion (e.g., I^−^, Br^−^, Cl^−^). The parameter *n* corresponds to the number of inorganic octahedral layers sandwiched between organic spacer layers (Figure 1a) [3]. These layers create strong quantum and dielectric confinement effects that regulate electronic properties and energy landscapes.

### 2.1. Structural and Compositional Features

The reduced dimensionality of inorganic slabs induces strong quantum confinement in quasi-2D perovskites, limiting charge carriers within the inorganic slabs and boosting radiative recombination. As *n* increases, the confinement effects diminish, leading to a redshift in the photoluminescence (PL) emission wavelength (Figure 1b) [3]. For instance, in *n* = 1 structures, the exciton binding energy (*E*_b_) can exceed 200 meV due to the strong confinement, while *E*_b_ decreases to values comparable to 3D perovskites (~10–50 meV) for *n* ≥ 4 [35]. This tunability enables precise control over emission wavelengths, ranging from the deep blue (*n* = 1–2) to the near-infrared (*n* ≥ 5) regions, making quasi-2D perovskites highly versatile for light-emitting applications.

### 2.2. Carrier Recombination Dynamics and PLQY Optimization

The recombination kinetics in quasi-2D perovskites are governed by a complex interplay of radiative and nonradiative pathways. Transient absorption (TA) and time-resolved photoluminescence (TRPL) studies have revealed that the dominant recombination mechanism critically depends on the *n*-value and excitation density. At low carrier densities (*N* < 10^16^ cm^−3^), excitonic recombination dominates in low-*n* phases (*n* = 1–3), characterized by a high radiative rate constant (*k*_1,exciton_) due to large *E*_b_. In contrast, high-*n* phases (*n* ≥ 4) exhibit free-carrier-dominated recombination, with a significant contribution from bimolecular processes (*k*_2_) [35]. Auger recombination (*k*_3_), however, becomes prominent at high excitation densities (*N* > 10^18^ cm^−3^), leading to efficiency roll-off in LEDs [36]. These findings show how important it is to find the critical balance between phase distribution and excitation conditions to optimize PLQYs.

Achieving high PLQYs (>80%) in quasi-2D perovskites requires efficient electroluminescence under low excitation levels. The PLQY is determined by the competition between radiative (*k*_rad_ = *k*_1,exciton_ + *k*_2_) and nonradiative (*k*_non-rad_ = *k*_1,trap_ + *k*_3_) recombination rates. Defect passivation strategies, such as introducing Lewis base additives (e.g., phosphine oxides or zwitterionic molecules), effectively suppress *k*_1,trap_ by coordinating under-coordinated Pb^2+^ ions and passivating halide vacancies [37,38,39]. For example, Yu et al. incorporated a dual passivation additive, diphenylphosphoramide (DPPA), into perovskite thin films, where the phosphine oxide group coordinates with unsaturated lead ions, while the amino group forms hydrogen bonds with adjacent halide ions to inhibit migration [38]. The blue quasi-2D PeLED based on DPPA-modified perovskite thin film achieved an external quantum efficiency of 12.31%. In addition, due to the decrease in defect density and suppression of ion migration in perovskite films, the device’s lifespan has been extended by 32%, and the spectrum is more stable. Similarly, Li et al. reported that multi-fluorophosphate additives reduce trap densities by 60%, enabling near-unity PLQYs in green-emitting films [39].

### 2.3. Energy Funneling Mechanisms

A hallmark of quasi-2D perovskites is their intrinsic energy funneling process, where photogenerated carriers rapidly transfer from wide-bandgap low-*n* phases to narrow-bandgap high-*n* phases. Time-resolved TA spectra shows that this transfer arises from electronic energy-level differences across the multi-phase film [40]. Despite these advances, challenges persist in achieving uniform phase control and mitigating Auger recombination. The trade-off between energy funneling efficiency and quantum confinement (*n*-value reduction for blue emission) remains a critical bottleneck. Recent efforts in dual-cation engineering (e.g., Rb^+^/Cs^+^ co-doping) and halogen alloying show promise in stabilizing blue emission while maintaining efficient energy transfer [41,42]. Additionally, advanced characterization techniques, such as ultrafast spectroscopy and in situ grazing-incidence wide-angle X-ray scattering (GIWAXS), provide deeper insights into crystallization kinetics and phase evolution, guiding the design of next-generation quasi-2D perovskites [39,43].

In summary, quasi-2D perovskites offer a versatile platform for high-performance LEDs due to their tunable optoelectronic properties and intrinsic energy-funneling mechanisms. Further research on defect passivation, phase engineering, and recombination dynamics is essential to address existing challenges and unlock their full potential in optoelectronics.

## 3. Energy Transfer Strategies in Quasi-2D PeLEDs

Quasi-2D PeLED devices rely on heterostructures comprising organic and inorganic layers with distinct electronic properties. Tailoring the electronic properties of each layer and ensuring precise energy-level alignment within the device enable efficient charge injection and radiative recombination, thereby enhancing device performance. As shown in Figure 2, the highest occupied molecular orbital (HOMO) represents the highest energy level at which electrons can exist within the valence band of organic materials, governing hole injection and transport at the anode. Conversely, the lowest unoccupied molecular orbital (LUMO) represents the lowest energy level at which an electron can be found in the conduction band of organic materials. LUMO corresponds to the energy level of electrons and is essential for the injection and transport of electrons from the anode. The HOMO energy level of HTL materials should be the same as or slightly higher than the valence band maximum (VBM) of the perovskite used in the device. This procedure will make sure that the holes are moved to the emitter layer. To stop the flow of electrons, the LUMO energy level of the HTL should be higher than the conduction band minimum (CBM) of perovskite. On the contrary, in ETL materials, the LUMO energy level should be aligned with or slightly lower than the CBM of perovskite, while the HOMO energy level should be lower than the VBM. This arrangement is critical for keeping hole transport from the emission layer to a minimum, while promoting injected electrons to move efficiently. Accurate energy-level matching in HTL and ETL materials is crucial for ensuring effective charge injection, transport, and confinement. They play an important role in improving the overall functionality and effectiveness of perovskite-based electronic devices [44].

Efficient quasi-2D PeLEDs require well-designed energy transfer mechanisms across material interfaces and device architectures to achieve high electroluminescence efficiency. These strategies are broadly categorized into intramaterial energy funneling (governed by phase composition and additive engineering) and device-level interfacial engineering (focused on charge transport layer optimization and energy-level alignment). Below, we dissect these approaches, emphasizing their interplay and impact on device performance.

### 3.1. Intra-Material Energy Funneling: Phase and Additive Engineering

#### 3.1.1. Phase Composition and Distribution Control

The energy funneling process in quasi-2D perovskites is intrinsically linked to the phase distribution of low-*n* (wide-bandgap) and high-*n* (narrow-bandgap) domains. Through rationally adjusting organic cation ratios and solvent selection, researchers can modulate the *n*-value distribution to enhance energy transfer efficiency. For instance, Sargent et al. demonstrated that films with engineered phase distributions (*n* = 3 and *n* = 5) exhibit accelerated carrier migration from low-*n* to high-*n* phases, boosting PLQYs by over 30% (Figure 3a,b) [40]. Guo et al. replaced some PEA^+^ with cations with weaker assembly ability (DPPA^+^ and TPMA^+^) to regulate nucleation and phase distribution, as shown in Figure 3c,d. The signal of the low *n* (especially *n* = 2) phase in the DPPA^+^ group was significantly weakened, indicating that phase separation was inhibited. In the TA spectrum of the TPMA film, only one large *n* (*n* ≥ 5) phase was observed (Figure 3c,d). In addition, studies have shown that grain orientation can also affect the energy transfer rate. This optimization minimizes carrier trapping in low-*n* phases, a critical bottleneck for blue emission [45]. Lei et al. further showed that choosing the right solvent is important because films made with N-methyl-2-pyrrolidone (NMP) transfer energy faster than films made with dimethyl sulfoxide (DMSO), because the random phase orientation is slowed down and the crystallinity is improved (Figure 3e,f) [43]. In DMSO samples, during initial excitation, the *n* = 2 and *n* = 3 phases exhibit similar relaxation with decay time constants (≈0.3 ps), while the *n* ≥ 4 phases show a faster rise time (0.2 ps), indicating the accumulation of carriers, followed by slower decay (0.4 ps). In addition, the rise time of the 3D domain is 0.36 ps, which further supports the idea that the accumulation of excitons in the 3D domain is dominated by energy transfer from a high bandgap (low *n* phase) to a low bandgap (high *n* phase) within the same time scale. The thin film made of NMP shows a similar trend, while the signal of *n* ≥ 4 phases is slightly invisible. It should be noted that the rise time of the 3D domain in the NMP sample is about 0.13 ps, significantly faster than the observed 0.36 ps in the DMSO sample, indicating that the carriers in the NMP sample relax to the 3D domain faster than in the DMSO sample. This faster energy transfer rate is due to the highly oriented nanocrystalline structure provided by the NMP solvent. The energy transfer from the 2D domain to the 3D domain is more likely due to Förster resonance energy transfer (FRET), which depends on the overlap of donor emission and acceptor absorption spectra, transition dipole orientation, and donor acceptor separation. The oriented crystals of NMP samples have higher density and smaller spacing between domains. Therefore, the reduction of donor acceptor spacing improves energy transfer efficiency. Such studies underscore the importance of crystallization dynamics in defining the energy landscape. These findings highlight that phase distribution control is not only pivotal for energy funneling but also critical for suppressing nonradiative losses in low-*n* phases.

#### 3.1.2. Additive Engineering for Defect Passivation

Additives play a dual role in enhancing energy funneling by passivating defects and regulating phase growth. Lewis base molecules (e.g., phosphine oxides, zwitterions) effectively coordinate undercoordinated Pb^2+^ ions and suppress halide migration. For example, Chen et al. managed the growth of particles in perovskite films by using the multifunctional organic molecule 2-amino-1,3-propanediol (APDO), which improved the efficiency of pore injection and optimized charge balance, as shown in Figure 4a [37]. The introduction of APDO molecules significantly improves the wettability of the HTL layer, promotes the growth of perovskite crystals during the spin coating process, effectively reduces trap states at the perovskite interface, and thus improves charge transfer. Li et al. added FDPP to perovskite, where the P=O bond can interact with Pb^2+^ in the perovskite precursor, effectively regulating the crystallization process and reducing the formation of defect centers (Figure 4b). In addition, F^−^ can bind with organic cations, slowing down the crystallization rate of perovskite crystals and facilitating the formation of high-quality RP perovskite thin films [39]. They obtained decreased low-n phase and higher PLQY films after adding FDPP. The use of the space-charge limited current (SCLC) method reflects the reduction of defect state density. The first stage corresponds to the ohmic contact region (*n* = 1), and the second stage represents the defect filling region, corresponding to the voltage at the defect filling limit (VTFL). The third stage corresponds to the defect-free space-charge limited current (*n* = 3). The decrease in the starting voltage of the trap filling area corresponding to VTFL from 1.94 V to 1.18 V proves the reduction in the defect density of the states. Furthermore, zwitterionic 3-(benzydimethylammonio) propanesulfonate (3-BAS) was shown to passivate surface defects while inhibiting halide ion migration in blue-emitting systems, elevating PLQYs from 45% to 82% (Figure 4c) [46]. Additives like TEAC (triethylammonium chloride) also narrow phase distributions by suppressing high-*n* (*n* ≥ 5) domains, enabling spectral blue shifts and prolonged carrier lifetimes (Figure 4d) [47]. These advancements highlight additive engineering as a versatile tool for balancing energy transfer and defect suppression.

### 3.2. Device-Level Interfacial Engineering: Charge Injection and Energy-Level Alignment

The interface is a critical region for carrier transport and exciton dissociation, which has a significant impact on the performance of light-emitting diodes. In quasi 2D PeLEDs, interfacial energy transfer strategies focus on optimizing bottom and top interfaces as well as dual-interface modulation.

#### 3.2.1. Bottom Interface Optimization

The perovskite/hole transport layer (HTL) interface, critical for hole injection efficiency and interfacial recombination, can be optimized by modifying the HTL with functional interlayers or molecules to address energy level mismatches and defect states. Jiang et al. inserted a BCPO (Bis-4-(N-carbazolyl)phenyl)phenylphosphine oxide) interlayer between the PEDOT:PSS and the perovskite film, enhancing hole mobility and film flatness, which formed an interface dipole to allow heterojunction band bending and obtain enhanced electron injection. The final EQE improved by 40% (Figure 5a,b) [48]. Zhao et al. showed that triphenylphosphine oxide (TPPO) at the HTL interface cancels out halide vacancies by coordinating P = O − Pb^2+^ and balancing charge injection. It achieved a record EQE of 18.2% in green PeLEDs at that time [49]. Zhang et al. further utilized GASCN (guanidinium thiocyanate) to suppress low-*n* (*n* = 1–2) phase formation, ensuring efficient energy funneling and carrier mobility (Figure 5c,d) [50].

#### 3.2.2. Top Interface Optimization

Similarly, the perovskite/electron transport layer (ETL) interface is vital for electron injection and exciton confinement, and its optimization can significantly enhance device performance. Zhu et al. reported that high-triplet-energy ETL materials (e.g., DPEPO) create energy barriers at the interface. Restricted to a high triplet energy level, a large number of triplet excitons are bound to the perovskite layer instead of being transferred to the ETL behind, thereby improving the utilization efficiency of excitons. This stops nonradiative recombination and makes sky-blue PeLEDs more efficient at radiative conversion (Figure 6a,b) [51]. Park et al. introduced TOPO (trioctylphosphine oxide) as a passivation layer on perovskite surfaces, reducing interfacial trap densities and improving stacking (Figure 6c,d) [52]. Such strategies highlight the need for ETL materials with tailored energy levels and defect-passivating capabilities.

#### 3.2.3. Dual-Interface Modulation: Insulator–Perovskite–Insulator (IPI) Structures

Dual-interface modulation is a strategic approach to simultaneously optimize charge injection and confinement at both the top and bottom interfaces of perovskite devices. This methodology addresses critical challenges in PeLEDs, such as unbalanced carrier injection, interfacial recombination losses, and exciton quenching, by means of engineering interfacial energy barriers and defect passivation. While conventional designs focus on single-interface optimization (e.g., HTL or ETL modification), dual-interface modulation enables holistic control of carrier dynamics, enhancing both efficiency and stability. A prominent example of this strategy is the insulator–perovskite–insulator (IPI) architecture, which replaces traditional charge transport layers (CTLs) with ultrathin insulating LiF films. These insulating layers facilitate carrier injection via tunneling effect while suppressing leakage currents and exciton quenching (Figure 7c). For instance, Shi et al. optimized LiF thickness in IPI-structured PeLEDs, demonstrating that a 4 nm LiF layer balances tunneling efficiency and carrier confinement, achieving an EQE of 9.0% in PeLED devices (Figure 7a,b) [53]. The insulating barriers decouple energy-level alignment requirements from charge injection mechanisms, enabling simplified device design and broader material compatibility [54,55]. The IPI architecture exemplifies the advantages of dual-interface modulation: simplified fabrication (eliminating CTLs), reduced production costs, and enhanced hole transport efficiency. By blocking electron leakage while enabling unidirectional hole injection, IPI structures mitigate efficiency roll-off at high current densities. However, challenges remain in optimizing tunneling efficiency for thick insulating layers and ensuring long-term stability under operational bias. Future research could explore alternative insulating materials (e.g., high-κ dielectrics) or hybrid designs integrating selective contacts with dual-interface engineering. This “CTL-free” paradigm not only advances the fundamental understanding of interfacial physics but also paves the way for scalable manufacturing of high-performance PeLEDs.

### 3.3. Supplementary Strategies for Enhanced Performance

#### 3.3.1. Carrier Management

By adjusting the materials of the charge-carrier injection and transport layers, and adding a charge-carrier blocking layer, the thickness of the perovskite layer, the injection, transport, and recombination processes of charge carriers can be better controlled. This approach makes the device more stable and increases its luminescence. For example, zinc oxide (ZnO) is often used as an ETL material because it has many electrons that can move around, the right energy levels, is easy to deposit, and is stable [56]. However, the high energy level of zinc oxide often leads to higher turn-on voltages and reduces the electron injection efficiency of the device [57]. To alleviate this concern, Qasim et al. used calcium-doped zinc oxide (CZO) nanoparticles (NPs) as the ETL [58]. Adding calcium to zinc oxide controls its energy levels, which helps align the conduction energy levels from the cathode to the perovskite emission layer (EML). This lowers the device’s turn-on voltage and makes electron injection more efficient. In addition, organic compounds can serve as dopants in addition to inorganic ones. Zhang et al. added 2,2″,2″(1,3,5-phenyltriphenyl) tris [1-phenyl-1benzimidazole] (TPBi) (9,9-bis (3′-(N, Ndimethylamino) propyl)-2,7-fluorene)–alt-2,7-(9,9-dioctylfluorene)] (PFN) to the polyfluoroene polymer [59]. As shown in Figure 8a, this adjustment resulted in the customized band structure of PFN, effectively reducing the HOMO of PFN, thereby enhancing hole injection blockage and significantly improving device performance. Wang et al. introduced TFB into PEDOT:PSS to promote hole injection while ensuring that the film morphology was not damaged and that no additional defects were introduced [41]. They also performed a cascade-like energy alignment between PEDOT:PSS and MAPbBr_3_ thin films (Figure 8b).

#### 3.3.2. Photon Management

By designing photonic crystal structures and introducing light scattering or reflection layers, the extraction efficiency of photons can be improved, and the light output efficiency of the device can be enhanced. Compared to organic semiconductors, perovskite typically has a relatively high refractive index, but due to its narrow escape cone, its optical efficiency is lower [60]. A theoretical study by Mei et al. showed that the microcavity effect makes PeLEDs 51% more efficient at extracting light, which is much higher than the classical theoretical estimate of about 20% [61]. In addition to interference in microcavities, the improvement in efficiency is also attributed to the low surface plasmon loss and high-level dipole ratio under the Purcell effect. In microcavity PeLEDs, the Purcell factor of the horizontal dipole is three times that of the vertical dipole, resulting in a highly efficient horizontal dipole contributing 86% of the light output. Moreover, experiments have shown that microcavities can increase external quantum efficiency by 70%. Narrower spectra and shorter photoluminescence lifetimes were also observed. These phenomena are attributed to the simultaneous enhancement of light extraction efficiency and internal quantum efficiency.

#### 3.3.3. Stability Enhancement

At present, it has been found that factors that affect the lifetime of devices include thermal degradation, generation of defect states, moisture-induced degradation, and ion migration. Many studies have improved the stability and environmental adaptability of quasi-2D perovskite materials through material and interface engineering techniques, extending the service life of devices. For example, Zhao et al. emphasized that the thermal stability of HTL polymers in devices is a necessary characteristic for improving device stability and reducing EQE roll-off [42]. Figure 8c shows the relationship between EQE and current density for various HTL materials with different glass transition temperature (*T*_g_) properties. The results indicate that there is no significant difference between materials when the current density is less than 2 A/cm^2^. However, when the current density exceeds 2 A/cm^2^, Joule heating begins to increase and becomes a significant factor. At this point, the EQE roll-off trend, which is related to the *T*_g_ of the HTL material, becomes apparent. In addition, it indicates that using higher *T*_g_ polymer materials not only reduces the roll-off of EQE but also improves the lifespan of the device due to increased thermal stability. In addition, although the ability to spin-coat the luminescent layer is a key advantage of PeLEDs, the solvents in the perovskite precursor can damage the HTL during device fabrication, potentially leading to undesirable fluorescence quenching and reduced EQE. To solve this problem, Matsushima et al. added tetraethyl orthosilicate (TEOS), a crosslinked polymer additive, into a traditional poly (N-vinylcarbazole) (PVK) HTL [62]. After heating during the spin-coating process, these mixed polymers can form silicon oxide polymers (siloxanes), making the film insoluble in common polar solvents. The EQE of the obtained device is increased by 1.5 times compared to unmodified due to less damage to the HTL layer during the perovskite deposition process, which reduces the quenching effect. Ma et al. provided atomic-scale insights into the moisture-induced degradation of perovskite crystals, revealing a surface-dependent dissolution pathway driven by ion solvation accompanied by shape transformation. The surface passivation of halide ion ligands combined with hydrophobic polymers significantly alters the degradation trajectory, maintains the cubic morphology, and reduces the dissolution rate [63]. In addition, metal/halide ions will migrate under an electric field, causing more vacancy defects under high voltage, thereby affecting device stability. To address this issue, Zhang et al. employed a LiF/perovskite/LiF structure and a ZnS/ZnSe cascade electron transport layer [64]. The LiF connecting the perovskite emission layer and the ETL prevents fluorescence quenching at the interface, and the combination of ZnS and ZnSe can be used as a cascaded ETL to reduce the energy barrier of efficient electron injection from the metal cathode to the perovskite and block holes. The prepared PeLED exhibits excellent storage stability (maintaining 90% of the initial EQE after 264 h) and operational stability (with a half-life of approximately 255 h at an initial brightness of 120 cd/m^2^, Figure 8d). Park et al. adopted a strategy of introducing potassium thiocyanate (KSCN) into perovskite, which significantly improved device efficiency by capturing free halide ions through K^+^ [65].
Figure 8(**a**) Energy-level diagram of a device with PFN:TPBi and J-V curves of electron/hole-only devices. Reproduced with permission from ref. [59]. Adv. Opt. Mater.; published by Wiley, 2019. (**b**) Fermi energy-level change with TFB and the EQE-J curves of PeLED after adding TFB as HTL. Reproduced with permission from ref. [41]. Micromachines; published by MDPI, 2022. (**c**) EQE current density curves of different HTL materials. Reproduced with permission from ref. [42]. Nano Lett., published by ACS, 2023. (**d**) LiF/Perovskite/LiF/ZnS/ZnSe structure and operational lifetime (T_50_) of the (Cs_0.83_Rb_0.17_)_0.95_K_0.05_-based device with L_0_ of 120 cd/m^2^. Reproduced with permission from ref. [64]. Adv. Funct. Mater.; published by Wiley, 2020.
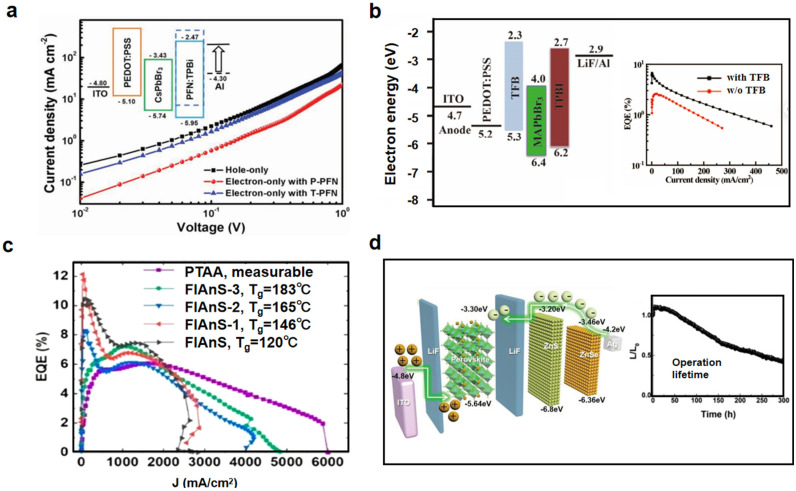


## 4. Challenges in Energy Transfer Research for Quasi-2D PeLEDs

Despite significant advancements in quasi-2D PeLEDs, there are still big problems that need to be solved before high-performance devices can be sold. These problems include spectral stability, efficiency roll-off, operational durability, and scalable fabrication. To solve them, we need solutions that come from different fields, such as material design, device engineering, and physical mechanisms.

### 4.1. Spectral Stability in Blue Emission

Blue-emitting quasi-2D PeLEDs have to choose between quantum confinement (required for short-wavelength emission) and energy funneling efficiency. Low-*n* phases (*n* = 1–2) exhibit strong confinement effects, but inefficient carrier transfer due to large exciton binding energies (*E*_b_ > 200 meV) [42]. Conversely, high-*n* phases (*n* ≥ 4) facilitate energy funneling but suffer from spectral redshift and phase heterogeneity [43]. Dual-cation engineering (e.g., Rb^+^/Cs^+^ co-doping) and halogen alloying (Cl^−^/Br^−^) have emerged as promising strategies to stabilize blue emission. Jiang et al. demonstrated that Rb^+^ doping in Cs-based quasi-2D perovskites suppresses halide migration and Ostwald ripening, achieving spectrally stable deep-blue emission at 454 nm [66]. Similarly, Gangishetty et al. utilized mixed halide systems with cationic surfactants to mitigate ion segregation, achieving EQEs exceeding 10% in blue PeLEDs [67]. However, the intrinsic instability of Cl^−^-containing perovskites under operational bias remains unresolved, demanding further exploration of organic cations with halogen-anchoring capabilities [68].

### 4.2. Efficiency Roll-Off at High Current Densities

Efficiency roll-off in quasi-2D PeLEDs primarily stems from Auger recombination (AR), which scales with the cube of carrier density (*k*_3_ ∝ *N*^3^). When the injection level is high (*N* > 10^18^ cm^−3^), Auger processes take over and break down PLQY quickly [41,42]. Furuhashi et al. found a complex relationship between the spacing of phenylalkylammonium and the potential factors involved in the two-dimensional alkyl perovskite AR process with increased alkyl chain length. The carrier density threshold for AR in long-chain phenylalkylammonium thin films is one order of magnitude higher, and the AR rate of long-chain interstitial cations is lower. The binding energy, exciton phonon coupling, and morphology of films were systematically studied using low-temperature absorption and temperature-dependent photoluminescence spectroscopy. Exciton phonon coupling plays an important role in regulating the rate of AR, while the contribution from morphology/defect related scattering is relatively small [69]. This basis correlates well with the observed trend of the measured AR rate constant, where PPA- and PBA-based films exhibit slower AR rates than PEA-based films. When comparing PPA and PBA, the AR rates are similar, even though PBA has a larger exciton–phonon coupling strength. This can be rationalized by the additional role played by defects/film morphology on the exciton diffusivity in PPA, thus impacting the efficiency of AR. Hence, in the PPA case, the exciton–phonon coupling strength, and the coupling of excitons to defects, as well as its film morphology, help to reduce the AR rates. However, in the PBA case, it is the large exciton–phonon coupling strength alone that mainly gives rise to the slower AR (Figure 9a,b). Jiang et al. further reduced the AR coefficient k_3_ by designing a low-Eb quasi-2D perovskite using polar p-F-PEA^+^ cations, achieving stable electroluminescence at a current density of up to 100 mA/cm^2^ [70]. Bi et al. suppressed AR in quantum dot systems by replacing the initial organic ligand with an inorganic ligand. Inorganic ligands act as “capacitors”, reducing charge accumulation and lowering the exciton binding energy of perovskite, thereby inhibiting AR and improving device efficiency [71]. Interface ion migration, Joule heating, and defect diffusion hinder long-term stability. Mobile ions can aggregate and penetrate the charge transport layer at the interface. The bending of the transport energy level and the decrease in charge transfer capacity exacerbate the unevenness of the charge injection and trigger a decrease in EQE. Joule heating can also exacerbate the decay of EQE, which can be attributed to ion processes, such as quenching or thermal activation [71,72]. Li et al. alleviated this problem by doping trifluoroacetate ions to decouple the electron hole wave function, delaying AR and reducing efficiency by 40% [39,72]. Li et al. demonstrated that doping electron-withdrawing trifluoroacetate anions into perovskite emitters leads to delayed AR due to the decoupling of electron hole wave functions. From Figure 9c–e, we know that it reduces the AR coefficient k_3_, limits the threshold for carrier density at which AR occurs, and thus reduces AR. Trifluoroacetate anions can also alter crystallization kinetics, inhibit halide migration, promote charge injection balance, and improve the tolerance of perovskite under high pressure [72].

### 4.3. Operational Stability and Degradation Mechanisms

Long-term stability is hindered by interfacial ion migration, Joule heating, and defect proliferation [73,74,75,76]. Ion migration at perovskite/charge transport layer interfaces induces electric field screening and phase segregation, accelerating device degradation [77,78]. Also, engineering the grain orientation has been shown to enhance carrier transport and reduce Joule heating, mitigating irreversible damage to functional layers [78]. Advanced encapsulation techniques and moisture-resistant perovskite formulations are urgently needed to extend device lifetimes beyond 1000 h under ambient conditions.

### 4.4. Scalable Fabrication and Commercialization

Performance degradation in large-area PeLEDs hinders the transition from lab-scale devices to industrial applications. Solution-processed films lose a lot of efficiency at scales larger than 10 cm^2^ because the phases are not spread out evenly, and there are groups of defects. But display applications require large-area devices. In response to this demand, Liu et al. used inkjet printing to prepare perovskite luminescent layers [79]. They first used a double-hole transport layer and a wetting interface layer to improve surface wettability, allowing printed perovskite droplets to form a continuous wet film. They controlled the fluid dynamics and evaporation kinetics of the perovskite wet layer through solvent engineering to suppress the coffee ring effect. Finally, uniform perovskite thin films were obtained on flexible substrates with different perovskite compositions. The peak EQE of the inkjet-printed PeLED reached 14.3%. The large-area flexible PeLED (4 × 7 cm^2^) also exhibited very uniform emission. Muratov et al. deposited an electron transport layer using a slit coating method, which replaced vapor deposition and reduced preparation costs [80]. At the same time, a more uniform and continuous thin film was formed, which lowered the device’s turn-on voltage. In addition, vacuum deposition is also a promising method for integrating PeLEDs into smart displays, as it has high manufacturability and is easy to pixelate. However, achieving spatially constrained grains with optimized crystals in vacuum-deposited perovskite still poses challenges. Zhang et al. proposed a three-source co-evaporation strategy, introducing MABr to form the MA_x_Cs_1−x_PbBr_3_ structure, which possesses spatial confinement of charge carriers and defect suppression [81]. This method increases the EQE of PeLEDs by nearly 10 times. But as the effective area increases, the above-mentioned problems at the perovskite interface become more apparent; the nonradiative recombination rate greatly increases, which leads to the inability of these technologies to achieve better efficiency on larger display devices.

## 5. Conclusions and Perspectives

Quasi-2D PeLEDs have emerged as a revolutionary platform for next-generation optoelectronics, offering tunable emission spectra, high PLQYs, and solution processability. The smart engineering of energy transfer processes, especially at material interfaces and device heterojunctions, is a key part of their progress. Based on our analysis of the PeLEDs issue, we believe that the core is the interface problem. This review systematically dissects the interplay between material design, interfacial dynamics, and device architecture, highlighting the problems that exist at the interface of perovskite and corresponding solutions. Based on these interface engineering solutions, we have the outlook described below for the future of PeLEDs.

### 5.1. Material Innovation and Design

Organic spacer cations and phase-distribution engineering make it possible for quasi-2D perovskites to have a lot of different structural options. This has made it possible to precisely control energy landscapes. Dual-cation systems (e.g., Rb^+^/Cs^+^ co-doping) and halogen alloying (e.g., Cl^−^/Br^−^) have shown promise in stabilizing blue emission by reducing halide migration and phase heterogeneity. Additive engineering, such as zwitterionic molecules and Lewis bases, further enhances radiative recombination by passivating undercoordinated Pb^2+^ and suppressing ion migration. Future research should focus on designing multifunctional organic cations that synergistically anchor halides, regulate crystallization kinetics, and enhance exciton confinement, thereby bridging the efficiency–stability gap in blue emission.

### 5.2. Device Architecture and Scalability

Interfacial engineering remains pivotal for balancing charge injection and minimizing nonradiative losses. The insulator–perovskite–insulator (IPI) structure exemplifies a paradigm shift because it allows carrier tunneling while blocking leakage currents. However, scalable fabrication of large-area PeLEDs demands defect-tolerant deposition techniques (e.g., slot-die coating) and thermally stable transport layers. Innovations in photon management, such as microcavity design, could further enhance light extraction efficiency by addressing the inherent trade-off between high refractive indices and optical out-coupling.

### 5.3. Mechanistic Insights and Stability

To prevent efficiency loss and degradation, it is necessary to have a deeper understanding of recombination dynamics, especially Auger processes and ion migration. Using advanced tools for characterizing materials (e.g., ultrafast spectroscopy and in situ GIWAXS) along with mathematical modeling will help us understand how carriers and photons interact and how phases change under operational stress. For example, methods to separate electron–hole wavefunctions or lower the density of carriers in a small area have been shown to stop Auger recombination. At the same time, encapsulation technologies and moisture-resistant formulations must evolve to extend device lifetimes beyond 1000 h under ambient conditions.

### 5.4. Eco-Friendly PeLEDs

For environmental considerations, selecting suitable B-site metal ions to replace traditional toxic lead is becoming a focus of sustainable development. There have been reports on tin-based PeLEDs, but due to the challenge of obtaining high-quality tin perovskite films, the development of tin-based PeLEDs has been relatively slow. Zhang et al. used a steam-assisted spin-coating method to achieve high-quality tin perovskite and high-efficiency LEDs. The results indicate that solvent vapor can cause in situ recrystallization of tin perovskite during the film formation process, significantly improving the quality of crystallization and reducing defects. Further introduction of antioxidant additives to suppress the oxidation of Sn^2+^ increase the photoluminescence quantum efficiency to about 30%, which is approximately four times higher than the baseline. The final preparation of tin-based PeLEDs achieved a peak EQE of 5.3% [82]. We look forward to better applications of interface engineering in eco-friendly devices.

### 5.5. Machine Learning to Improve PeLED Performance

New materials often exhibit unexpectedly beneficial properties, surpassing simple similar materials. The increasing trend of material complexity requires a systematic strategy to explore multi-element “multi-component engineering”. Perovskite has excellent material versatility and is suitable for high-throughput screening, machine learning, and data mining. Regarding the various issues raised above, we believe that the establishment of an effective mathematical model and the use of machine learning methods for screening and optimization will play important roles in promoting the application of PeLEDs.

In conclusion, quasi-2D PeLEDs stand at the forefront of emissive technologies, with their unique energy transfer mechanisms offering unparalleled opportunities for high-efficiency lighting and displays. By addressing existing bottlenecks using new materials, better devices, and focusing on how things work, this field is poised to redefine the future of optoelectronics.

## Figures and Tables

**Figure 1 nanomaterials-15-00592-f001:**
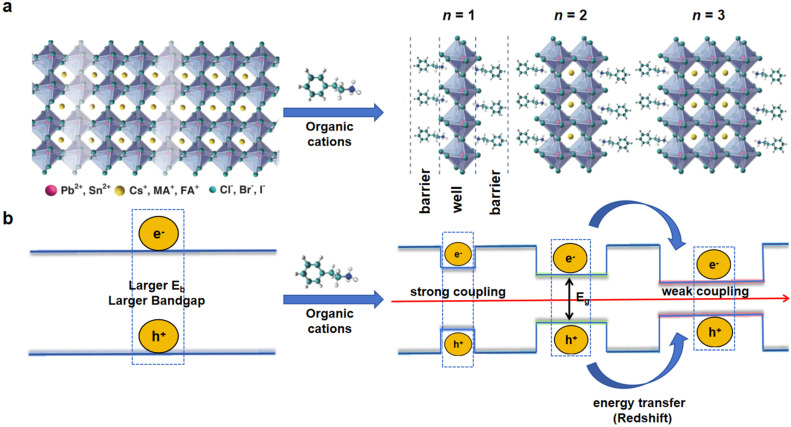
(**a**) Schematic representation of a quasi-2D perovskite, which can be obtained by slicing the 3D perovskite along the <100> crystallographic direction with different *n* values. Reproduced with permission from [3], Light Sci. Appl.; published by Nature, 2021. (**b**) Electronic properties of quasi-2D perovskites, which are determined by the degree of quantum and dielectric confinement effects.

**Figure 2 nanomaterials-15-00592-f002:**
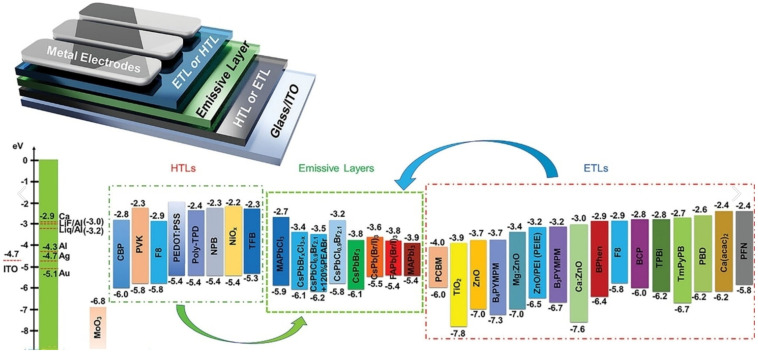
A general PeLED device structure is shown in this figure. It includes energy-level diagrams for HTL, ETL, and emissive perovskite materials that are commonly used. For inorganic (or hybrid) semiconductors, such as metal oxides and perovskites, the conduction band minima (CBM) and valence band maxima (VBM) are the upper and lower energy levels, respectively. For organic materials, the upper and lower levels represent the lowest unoccupied molecular orbital (LUMO) and highest occupied molecular orbital (HOMO) energies, respectively. The red dashed lines on the left depict the Fermi levels of typical metallic electrode materials and metal/insulator interfaces. Reproduced with permission from ref. [44], Chem. Eur. J.; published by Wiley, 2024.

**Figure 3 nanomaterials-15-00592-f003:**
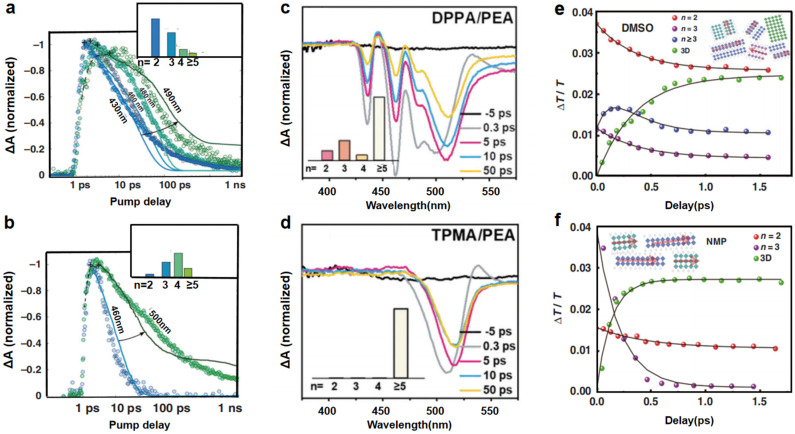
Time-dependent TA spectral traces for (**a**) *n* = 3 perovskite film with an engineered energy landscape and (**b**) *n* = 5 film with a graded energy landscape. Reproduced with permission from ref. [40]. Nano Lett., published by ACS, 2017. Transient absorption spectra of Q-2D perovskites with different spacer cations (DPPA/TPMA) substitution. The inset in (**c**,**d**) represents the relative contributions of different n phases in each film according to the integral intensity of GSB in the corresponding TA spectra. Reproduced with permission from ref. [45]. Adv. Mater.; published by Wiley, 2023. Time-dependent TA spectral traces of different quasi-2D perovskite films obtained from (**e**) DMSO and (**f**) NMP solvents. Reproduced with permission from ref. [43]. Adv. Mater.; published by Wiley, 2020.

**Figure 4 nanomaterials-15-00592-f004:**
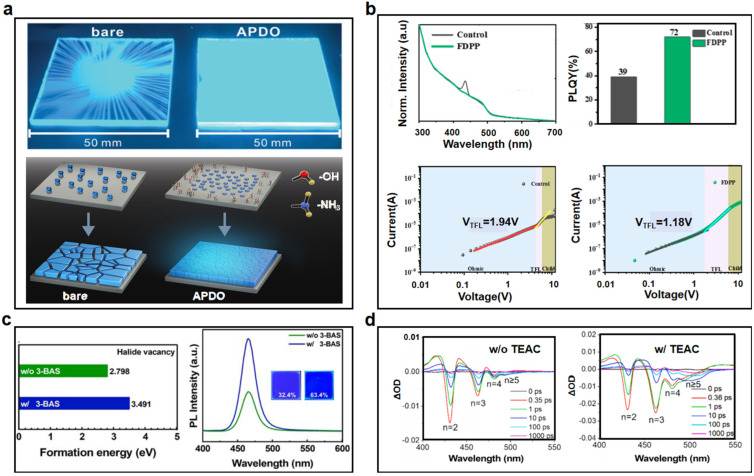
(**a**) Schematic crystal structures of quasi-2D perovskites with APDO and their surface appearance. Reproduced with permission from ref. [37]. Adv. Sci.; published by Wiley, 2021. (**b**) Schematic crystal structures of quasi-2D perovskites and the surface passivation condition after adding FDPP. Reproduced with permission from ref. [39]. Micromachines; published by MDPI, 2024. (**c**) The vacancy energy and PL intensity of perovskite films for reference and 3-BAS-treated samples. Reproduced with permission from ref. [46]. Adv. Photon.; published by Wiley, 2024. (**d**) Fs-TA spectra at different timescales for pristine and TEAC-modified samples. Reproduced with permission from ref. [47]. Chem. Eng. J.; published by Elsevier, 2024.

**Figure 5 nanomaterials-15-00592-f005:**
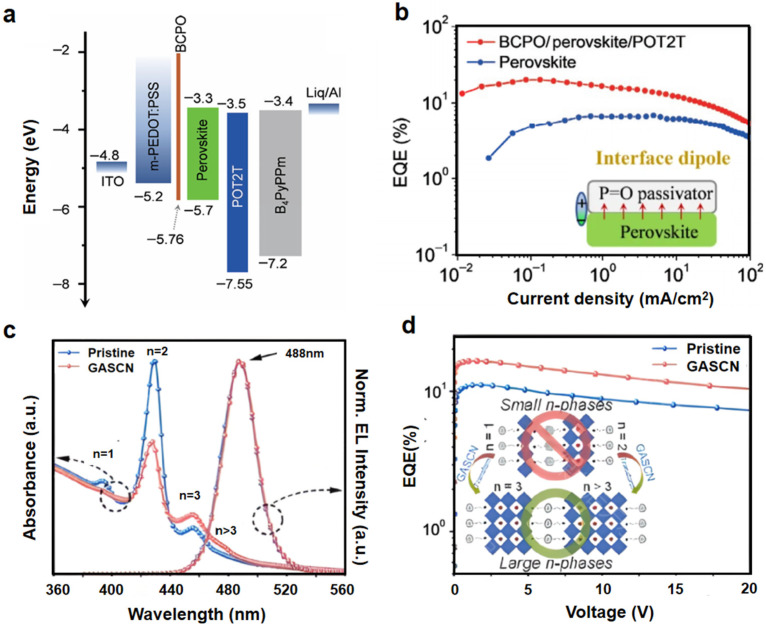
(**a**) Energy-level diagram of a device with BCPO. (**b**) EQE of the device with/without BCPO. Reproduced with permission from ref. [48]. Nano Res.; published by Springer Nature, 2022. (**c**) Absorbance of perovskite film with/without GASCN. (**d**) EQE of the final device with/without GASCN. Reproduced with permission from ref. [50]. Nano Energy; published by Elsevier, 2024.

**Figure 6 nanomaterials-15-00592-f006:**
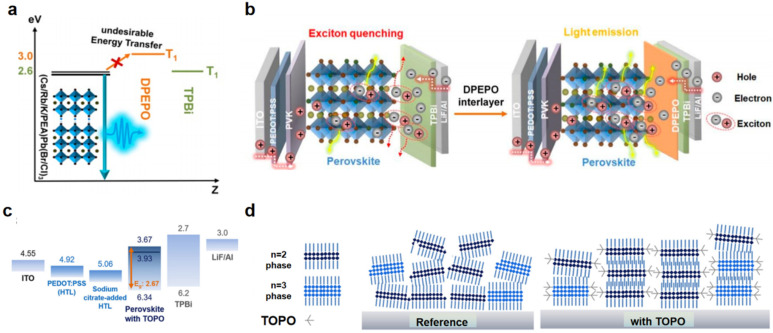
(**a**) Mechanism diagrams of the proposed DPEPO spacer layer inhibiting undesirable energy transfer. (**b**) The exciton interface recombination behavior of PeLEDs without and with a DPEPO spacer layer. Reproduced with permission from ref. [51]. J. Phys. Chem. Lett., published by ACS, 2021. (**c**) Energy-level diagram of the complete EL devices. (**d**) The schematics showing the stacking of low-dimensional phases of the reference and with TOPO. Reproduced with permission from ref. [52]. ACS App. Mater. Interfaces, published by ACS, 2024.

**Figure 7 nanomaterials-15-00592-f007:**
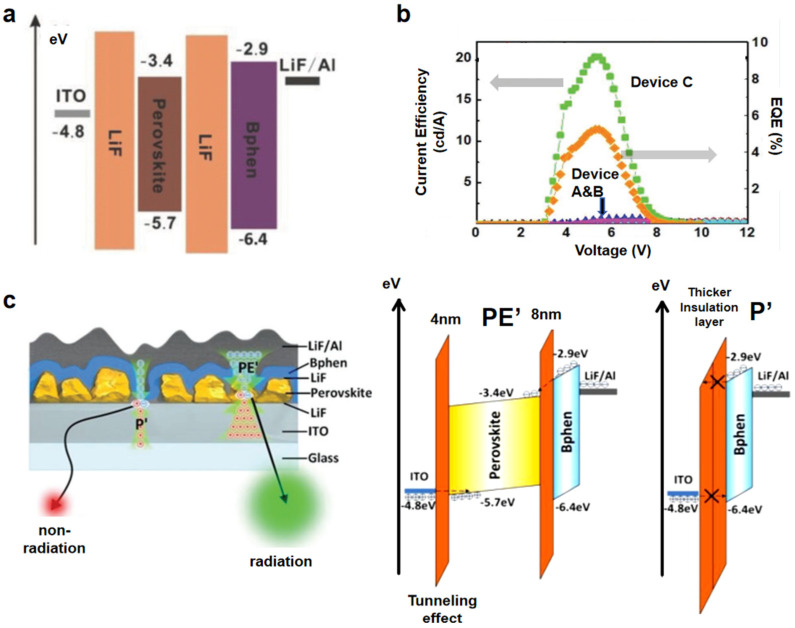
(**a**) Energy−level diagram of different layers of IPI-structured PeLEDs. (**b**) CE-V and EQE-V curves of Device A (ITO/PEDOT:PSS/perovskite/Bphen/LiF/Al), Device B (ITO/PEDOT:PSS/PVK/perovskite/Bphen/LiF/Al), and Device C (ITO/LiF/PVK/perovskite/LiF/Bphen/LiF/Al). (**c**) Device cross-section diagram with P′ (pinhole) and PE″ (crystal) regions and their energy transfer diagram: the PE″ region has a thinner energy barrier for hole injection, while the P″ region has a thick LiF layer that prevents tunneling of charge and effectively prevents leakage current. Reproduced with permission from ref. [53]. Adv. Mater.; published by Wiley, 2018.

**Figure 9 nanomaterials-15-00592-f009:**
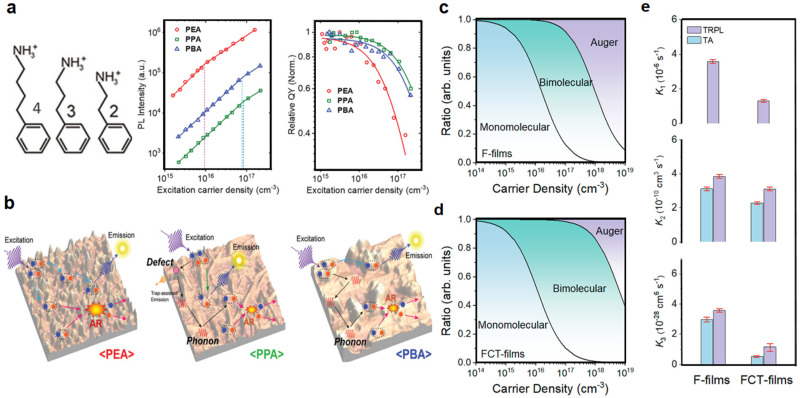
(**a**) Schematic image of organic spacer and PL, relative quantum yield (QY) for PEA, PPA, and PBA films. (**b**) Schematic of the photophysical processes for PEA, PPA, and PBA films. Reproduced with permission from ref. [70]. Nat. Commun.; published by Nature, 2021. Proportion of the recombination ratio of (**c**) F-and (**d**) FCT-film, respectively. (**e**) Derived values of recombination rate constant for the F- and FCT-films, where k_1_ was extracted from low-fluence TRPL spectra, and k_2_ and k_3_ were extracted from both TRPL and TA measurements. Reproduced with permission from ref. [72]. Nat. Commun.; published by Nature, 2025.

## Data Availability

Data are contained within the article.

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
