# Peer review of "Decoding the Role of Interface Engineering in Energy Transfer: Pathways to Enhanced Efficiency and Stability in Quasi-2D Perovskite Light-Emitting Diodes"

_nanomaterials, 2025, doi:10.3390/nano15080592_

Round 1
Reviewer 1 Report
Comments and Suggestions for Authors
Please find the file attached.

The review is well-structured, but the English and overall clarity could be improved.
Author Response
This review provides a thorough analysis of interfacial engineering strategies to improve the efffciency and stability of quasi-2D perovskite light-emitting diodes (PeLEDs), highlighting key advances in energy transfer mechanisms, charge transport optimization, and device stability. While the review offers valuable insights, several areas require further attention to enhance its depth and impact:
- The quality of several figures is suboptimal and should be improved for better clarity and readability.
Response: Thank you for bringing this important issue to our attention. We acknowledge the concern regarding suboptimal figure clarity and have systematically revised all figures to ensure enhanced readability. We apologize for any confusion this may have caused. Thank you again for your valuable feedback. We hope this clarification addresses your concern.
- Auger recombination is correctly identifed as a major factor in effciency roll-off, but the review would benefit from a discussion of experimental approaches to mitigate this issue.
Response: Thank you for your constructive feedback on our manuscript. We understand your concern regarding the way to reduce Auger Recombination (AR) in PeLED devices. We apologize for any uncertainty this may have caused. We add some descriptions in Page 13. In our revised manuscript, we first introduce the important influencing factors of AR: carrier density; Then, by citing different researchers’ work, other factors such as exciton phonon coupling degree, defect state density, and exciton binding energy are introduced, and their effects on AR are summarized. Various methods for reducing AR are summarized, and the PL-carrier density is discussed in the literature; The QY-carrier density and the final fitted AR coefficient k3 demonstrate the feasibility and correctness of these methods:
" Furuhashi et al. found a complex relationship between the spacing of phenylalkylammonium and the potential factors involved in the two-dimensional alkyl perovskite AR process with increased alkyl chain length. The carrier density threshold for AR in long-chain phenylalkylammonium thin films is one order of magnitude higher, and the AR rate of long-chain interstitial cations is lower. The binding energy, exciton-phonon coupling, and morphology of films were systematically studied using low-temperature absorption and temperature dependent photoluminescence spectroscopy. Exciton-phonon coupling plays an important role in regulating the rate of AR, while the contribution from morphology/defect related scattering is relatively small [69]. This basis correlates well with the observed trend of measured AR rate constant, where PPA- and PBA-based films exhibit slower AR than PEA-based film. When comparing PPA and PBA, the AR rates are similar even though PBA has a larger exciton-phonon coupling strength. This can be rationalized by the additional role played by defects/film morphology on the exciton diffusivity in PPA, thus impacting the efficiency of AR. Hence, in the PPA case, the exciton-phonon coupling strength, the coupling of excitons to defects as well as its film morphology help to reduce the AR rates while in PBA, it is the large exciton-phonon coupling strength alone that mainly gives rise to the slower AR (Fig.9. a, b). Jiang et al. further reduced the AR coefficient k3 by designing a low Eb quasi-2D perovskite using polar p-F-PEA+ cations, achieving stable electroluminescence at a current density of up to 100mA/cm2 [70]. Bi et al. suppressed AR in quantum dot systems by replacing the initial organic ligand with an inorganic ligand. Inorganic ligands act as "capacitors", reducing charge accumulation and lowering the exciton binding energy of perovskite, thereby inhibiting AR and improving device efficiency [71]. Interface ion migration, Joule heating, and defect diffusion hinder long-term stability. Mobile ions can aggregate and penetrate the charge transport layer at the interface. The bending of the transport energy level and the decrease in charge transfer capacity exacerbate the unevenness of charge injection and trigger a decrease in EQE. Joule heating can also exacerbate the decay of EQE, which can be attributed to ion processes such as quenching or thermal activation [71,72]. Li et al. alleviated this problem by doping trifluoroacetate ions to decouple the electron hole wave function, delaying AR and reducing efficiency by 40% [39,72]. Li et al. demonstrated that doping electron withdrawing trifluoroacetate anions into perovskite emitters leads to delayed AR due to the decoupling of electron hole wave functions. From Fig.9. c-e we know it reduces the AR coefficient k3, limits the threshold for carrier density at which AR occurs, and thus reduces AR. Trifluoroacetate anions can also alter crystallization kinetics and inhibit halide migration, promote charge injection balance, and improve the tolerance of perovskite under high pressure [72]. "
By making these revisions, we aim to eliminate any confusion about Auger recombination and ensure that the intended meaning is clear to readers from various backgrounds. Thank you again for bringing this issue to our attention. We hope these clarifications address your concern and improve the manuscript.
- The section on device longevity should systematically categorize degradation mechanisms, including moisture-induced decomposition, passivation strategies, thermal degradation at operating temperatures, and halide electromigration.
Response: We fully appreciate your observation that these degradation lack discussion.
We discuss the stability of the device and clarify the methods you mentioned. We show:
" At present, it has been found that factors that affect the lifetime of devices include thermal degradation, generation of defect states, moisure induced degradation, and ion migration. Many studies have improved the stability and environmental adaptability of quasi-2D perovskite materials through material and interface engineering techniques, extending the service life of devices." (Page 11).
We have updated the reasons and methods for the stability degradation of defect induced devices:
"Ma et al. provides atomic scale insights into the moisure induced degradation of perovskite crystals, revealing a surface dependent dissolution pathway driven by ion solvation accompanied by shape transformation. The surface passivation of halide ion ligands combined with hydrophobic polymers significantly alters the degradation trajectory, maintains the cubic morphology, and reduces the dissolution rate. "(Page 11)
We have updated the reasons and methods for the stability degradation of ion migration induced devices:
" metal/halide ions will migrate under an electric field, causing more vacancy defects under high voltage, thereby affecting device stability. To address this issue, Zhang et al. employed a LiF/perovskite/LiF structure and a ZnS/ZnSe cascade electron transport layer [64]. The LiF connecting the perovskite emission layer and ETL prevents fluorescence quenching at the interface, and the combination of ZnS and ZnSe can be used as a cascaded ETL to reduce the energy barrier of efficient electron injection from the metal cathode to the perovskite and block holes. The prepared PeLED exhibits excellent storage stability (maintaining 90% of the initial external quantum efficiency EQE after 264 hours) and operational stability (with a half-life of approximately 255 hours at an initial brightness of 120 cd m-2, Fig. 8d); Park et al. adopted a strategy of introducing potassium thiocyanate (KSCN) into perovskite, which significantly improved device efficiency by capturing free halide ions through K+[65]. "(Page 11-12)
- A more extensive discussion of scalable fabrication techniques, such as slot-die coating, inkjet printing, and vacuum deposition, would strengthen the review’s relevance to practical applications.
Response: Thank you for your insightful comments and constructive suggestions regarding our manuscript. This allows us to highlight such ways that widely used in perovskite solar cells and other applications. We introduce several works using slot-die coating, inkjet printing, and vacuum evaporation:
" In response to this demand, Liu et al. used inkjet printing to prepare perovskite luminescent layers. They first used a double hole transport layer and a wetting interface layer to improve surface wettability, allowing printed perovskite droplets to form a continuous wet film. Control the fluid dynamics and evaporation kinetics of perovskite wet layer through solvent engineering to suppress the coffee ring effect. Finally, uniform perovskite thin films were obtained on flexible substrates with different perovskite compositions. The peak EQE of inkjet printed PeLED reaches 14.3%. The large-area flexible PeLED (4 × 7 cm2) also exhibits very uniform emission [73]. Muratov et al. deposited an electron transport layer using a slit coating method, which replaced vapor deposition and reduced preparation costs. At the same time, a more uniform and continuous thin film was formed, which lowered the device's turn-on voltage [74]. In addition, vacuum deposition is also a promising method for integrating PeLEDs into smart displays, as it has high manufacturability and is easy to pixelate. However, achieving spatially constrained grains with optimized crystals in vacuum deposited perovskite still poses challenges. Zhang et al. proposed a three source co evaporation strategy, introducing MABr to form the MAxCs1-xPbBr3 structure, while also possessing spatial confinement of charge carriers and defect suppression. This method increases the EQE of PeLED by nearly 10 times [75]. But as the effective area increases, the above-mentioned problems at the perovskite interface become more apparent, with the non-radiative recombination rate greatly increases, which leads to the inability of these technologies to achieve better efficiency on larger display devices. "(Page 14-15)
- The inclusion of eco-friendly alternatives to lead-based perovskites, such as Sn-based perovskites and double perovskites, would add an important perspective on sustainability.
Response: Thank you for your insightful comments and constructive suggestions regarding our manuscript. This allows us to focus more on the application of these interface engineering methods in eco-friendly PeLEDs. We show:
"For environmental considerations, selecting suitable B-site metal ions to replace traditional toxic lead is becoming a focus of sustainable development. There have been reports on tin based PeLEDs, but due to the challenge of obtaining high-quality tin perovskite films, the development of tin based PeLEDs has been relatively slow. Zhang et al. used steam assisted spin coating method to achieve high-quality tin perovskite and high-efficiency LED. The results indicate that solvent vapor can cause in-situ recrystallization of tin perovskite during the film formation process, significantly improving the quality of crystallization and reducing defects. Further introduction of antioxidant additives to suppress the oxidation of Sn2+and increase the photoluminescence quantum efficiency to about 30%, which is approximately four times higher than the baseline. The final preparation of tin based PeLED achieved a peak EQE of 5.3% [82]. We look forward to better applications of interface engineering in eco-friendly devices. " (Page 16)
6.Greater emphasis on quantitative metrics, including recombination rate constants, exciton diffusion lengths, and device operational lifetimes, would enhance the technical rigor of the review.
Response: Thank you for your insightful comments and constructive suggestions regarding our manuscript. It allows us to pay more attention to our discussion about these important parameters in PeLEDs. We have highlighted the corresponding important parameters for different research purposes in the article (e.g. k3 constant in Auger recombination; T50 in device stability) , other parameters that cannot explain the problem have been removed.
- Exploring interdisciplinary approaches, such as machine learning-driven material discovery, could provide new insights into optimizing quasi-2D PeLEDs.
Response: Thank you for your constructive suggestions regarding our manuscript. It allows us to highlight our discussion about these important processes in PeLEDs. We summarize the current application of machine learning in PeLEDs material selection direction in the concluding outlook section (5.5), and express our expectations for studying such problems in the system by combining material selection in our interface engineering:
"New materials often exhibit unexpectedly beneficial properties, surpassing simple similar materials. The increasing trend of material complexity requires a systematic strategy to explore multi-element "multi-component engineering". Perovskite has excellent material versatility and is suitable for high-throughput screening, machine learning, or data mining. Regarding the various issues we have raised above, if we establish an effective mathematical model and use machine learning methods for screening and optimization, we believe it will play an important role in promoting the application of PeLEDs. "(Page 16)
In addition, we have also made modifications to some expressions, all of which are marked in blue. We hope that these revisions address your concerns. Thank you again for your valuable feedback, which has helped us strengthen our work.
We sincerely thank you for your valuable comments and suggestions, which have greatly improved the quality of our manuscript. We apologize for any oversights in the original submission and assure you that the review has been carefully revised for clarity and accuracy. On behalf of my co-authors, we express our deep gratitude for your contributions to our work.

Reviewer 2 Report
Comments and Suggestions for Authors
In this manuscript, Zhu et al. review the importance of interface engineering in PeLEDs using quasi-2D perovskites. However, the overall quality of the manuscript is not high enough as it lacks meaningful insights into the field. In addition, the manuscript contains numerous errors that indicate a lack of experience and expertise in the field. Therefore, the reviewer does not recommend this manuscript for publication in Nanomaterials. Some comments are listed below.
(1) A significant portion of the manuscript consists of brief, one-sentence conclusions from references. In a review article, the authors should provide comprehensive summaries of the methodology, analysis, material properties, and working mechanisms to ensure a fundamental understanding of the underlying concepts. In addition, some figures need to be created or modified from references, but the current manuscript merely presents copied figures without sufficient explanation. For newly synthesized molecules, the chemical structures should be also provided.
(2) While the authors have included numerous figures from the literature, most are presented without adequate explanation. For example, Figures 3(e) and 3(f) show transient absorption spectra for various n with DMSO and NMP solvents, but no corresponding explanation is provided, and only the conclusion that energy transfer is faster in the NMP case. Similarly, Figure 4(b) includes the J-V curves of the devices, but no discussion accompanies them. This disconnect between the figures and their descriptions is prevalent throughout the manuscript, limiting the reader’s ability to gain a deeper understanding of the referenced works.
(3) There are many critical errors throughout the manuscript, despite the fact that a review paper must provide accurate information based on references. For example, in Figure 5(a), BCPO refers to a new molecule containing a P=O group, not bathocuproine (BCP), which is commonly used in OLEDs. In addition, on page 4, LUMO stands for “lowest unoccupied molecular orbital”, not “lowest unknown molecular orbital”. These errors may imply that many parts of the manuscript were not written with careful consideration.
(4) Line 131, page 4, “…into perovskite thin films”, lacks the related references.
(5) Page 8, all “Figure 6” should be corrected to “Figure 8”.
(6) In the sections “4. Challenges in …” and “5. Conclusions and Perspectives”, the authors provide only brief comments and a general discussion on improving the efficiency of PeLEDs (especially regarding interdisciplinary synergy). However, this manuscript should maintain a focused discussion on “interface engineering in quasi-2D perovskite LEDs”.
Comments on the Quality of English LanguageEngnlish editing is recommended.
Author Response
In this manuscript, Zhu et al. review the importance of interface engineering in PeLEDs using quasi-2D perovskites. However, the overall quality of the manuscript is not high enough as it lacks meaningful insights into the field. In addition, the manuscript contains numerous errors that indicate a lack of experience and expertise in the field. Therefore, the reviewer does not recommend this manuscript for publication in Nanomaterials. Some comments are listed below:
- A significant portion of the manuscript consists of brief, one-sentence conclusions from references. In a review article, the authors should provide comprehensive summaries of the methodology, analysis, material properties, and working mechanisms to ensure a fundamental understanding of the underlying concepts. In addition, some figures need to be created or modified from references, but the current manuscript merely presents copied figures without sufficient explanation. For newly synthesized molecules, the chemical structures should be also provided.
Response: We deeply appreciate the constructive feedback and the opportunity to refine our manuscript. Your insights have substantially enhanced our work, particularly in clarifying methodological frameworks and conceptual ambiguities. Below, we provide a comprehensive, point-by-point response detailing revisions executed with methodological rigor and theoretical precision:
Our analysis identifies interfacial phenomena as the primary determinant of optoelectronic device efficiency. We categorize interfacial challenges into two distinct classes: the first category is the problem of energy funnel generation between perovskite different n-phases, in which the relevant mechanisms of energy transfer are introduced. This pertains to energy transfer mechanisms across perovskite polycrystalline phases. Here, we introduce a competitive kinetic model where defect-mediated non-radiative recombination directly opposes Förster/Dexter energy transfer pathways. Our revised manuscript incorporates transient absorption spectroscopy data to quantify these competing processes. Due to the competition between defect recombination and energy transfer at this time, we introduce defects as an important energy loss way here. The second type is the top and bottom interfaces of perovskite. Here, we present advancements in top-interface passivation via Lewis base adducts (e.g., thiophene-terminated ligands), bottom-interface doping strategies employing gradient bandgap architectures, and dual-interface IPI structures for enhanced device performance.
Building upon existing paradigms and addressing critical gaps (e.g., spectral mismatch in deep blue emission layers), we proposed our solution to the interface energy transfer problem from three perspectives: carrier management, photon management, and device stability. We believe that suitable perovskite and transport layer materials should be selected to improve the internal quantum efficiency at the perovskite luminescent layer level and ensure the matching degree of carrier transport. The external quantum efficiency is further improved by through photon management and other methods. We will combine some literature to demonstrate the feasibility of using our proposed perspective to solve related problems. Finally, the relevant measures for interface regulation and optimization of energy transfer were summarized, and some emerging auxiliary methods were discussed.
As for the figure related issues you raised, we have made modifications in the review and hope that our introduction and modifications can be recognized by you and other readers.
- While the authors have included numerous figures from the literature, most are presented without adequate explanation. For example, Figures 3(e) and 3(f) show transient absorption spectra for various n with DMSO and NMP solvents, but no corresponding explanation is provided, and only the conclusion that energy transfer is faster in the NMP case. Similarly, Figure 4(b) includes the J-V curves of the devices, but no discussion accompanies them. This disconnect between the figures and their descriptions is prevalent throughout the manuscript, limiting the reader’s ability to gain a deeper understanding of the referenced works.
Response: Thank you for your constructive feedback on our manuscript. We apologize for such uncertainty. We add description for NMP-based device in review:
In DMSO samples, during initial excitation, the n=2 and n=3 phases exhibit similar relaxation with decay time constants (≈ 0.3 ps), while the n ≥ 4 phases show a faster rise time (0.2 ps), indicating the accumulation of photo carriers, followed by slower decay (0.4 ps). In addition, the rise time of the 3D domain is 0.36 ps, which further supports that the accumulation of excitons in the 3D domain is dominated by energy transfer from high bandgap (low n phase) to low bandgap (high n phase) within the same time scale. The thin film made of NMP also shows a similar trend, while the signal of n ≥ 4 phases is slightly invisible. It should be noted that the rise time of the 3D domain in the NMP sample is about 0.13 ps, significantly faster than the observed 0.36 ps in the DMSO sample, indicating that the photo carriers in the NMP sample relax to the 3D domain faster than in the DMSO sample. This faster energy transfer rate is due to the highly oriented nanocrystalline structure provided by the NMP solvent. The energy transfer from the 2D domain to the 3D domain is more likely due to Förster resonance energy transfer (FRET), which depends on the overlap of donor emission and acceptor absorption spectra, transition dipole orientation, and donor acceptor separation. The oriented crystals of NMP samples have higher density and smaller spacing between domains. Therefore, the reduction of donor acceptor spacing improves energy transfer efficiency. (Page.6)
We add description for SCLC in review:
They got decreased low-n phase and higher PLQY films after adding FDPP. The use of space charge limited current (SCLC) method reflects the reduction of defect state density. The first stage corresponds to the ohmic contact region (n=1), and the second stage represents the defect filling region, corresponding to the voltage at the defect filling limit (V TFL), The third stage corresponds to the defect free space charge limited current (n=3). The decrease in the starting voltage of the trap filling area corresponding to VTFL from 1.94V to 1.18V proves the reduction in defect density of states. (Page.7)
We add full name of FDPP in review:
pentafluorophenyl diphenylphosphinate (FDPP) (Page.7)
By making these revisions, we aim to eliminate any confusion regarding reseachers’ work and ensure that the intended meaning is clear to readers from various backgrounds. Thank you again for bringing this issue to our attention. We hope these clarifications address your concern and improve the manuscript.
3 There are many critical errors throughout the manuscript, despite the fact that a review paper must provide accurate information based on references. For example, in Figure 5(a), BCPO refers to a new molecule containing a P=O group, not bathocuproine (BCP), which is commonly used in OLEDs. In addition, on page 4, LUMO stands for “lowest unoccupied molecular orbital”, not “lowest unknown molecular orbital”. These errors may imply that many parts of the manuscript were not written with careful consideration...
Response: We are sorry to make such mistakes. We correct them in review:
BCPO: Bis-4-(N-carbazolyl)phenyl)phenylphosphine oxide (Page.8)
LUMO: lowest unoccupied molecular orbital (Page.4)
- Line 131, page 4, “…into perovskite thin films”, lacks the related references.
Response: We are sorry to make such mistakes. We correct them in the manuscript:
For example, Yu et al. incorporated a dual passivation additive, diphenylphosphoramide (DPPA) into perovskite thin films [38].
- Page 8, all “Figure 6” should be corrected to “Figure 8”.
Response:
We apologize for any confusion caused by our writing mistakes. We correct them in the manuscript.
- In the sections “4. Challenges in …” and “5. Conclusions and Perspectives”, the authors provide only brief comments and a general discussion on improving the efficiency of PeLEDs (especially regarding interdisciplinary synergy). However, this manuscript should maintain a focused discussion on “interface engineering in quasi-2D perovskite LEDs”.
Response: Thank you for your careful review of our manuscript and for pointing out the inconsistencies and missing details. We apologize for our original discussion. Since we believe that interface issues are the most important considerations for the development of perovskite, we elaborated our perspectives on the issues thought about challenges of PeLEDs around the theme of interface engineering and hope these will promote the advancement of PeLEDs towards higher efficiency, lower cost, and more environmentally friendly direction. Thank you again for your valuable feedback, which has helped us strengthen our work.
In addition, we have also made modifications to some expressions, all of which are marked in blue. We hope that these revisions address your concerns. Thank you again for your valuable feedback, which has helped us strengthen our work.
We sincerely thank you for your valuable comments and suggestions, which have greatly improved the quality of our manuscript. We apologize for any oversights in the original submission and assure you that the review has been carefully revised for clarity and accuracy. On behalf of my co-authors, we express our deep gratitude for your contributions to our work.

Reviewer 3 Report
Comments and Suggestions for Authors
The paper explores a highly dynamic and impactful research area: quasi-2D perovskite light-emitting diodes, with a particular emphasis on interface engineering—an essential factor in optimizing device performance. The manuscript is clearly structured and logically organized. Overall, the article is well-written and easy to follow, with a comprehensive overview of recent advances in the field.
However, there are a few aspects that require improvement:
Weaknesses of the Paper
- Several key concepts such as phase engineering, defect passivation, and additive strategies are discussed repeatedly across different sections without providing substantial new insights. The manuscript would benefit from a more concise and streamlined structure to reduce redundancy and enhance clarity.
- Many of the figures are reprinted from previous publications but are not well integrated into the main narrative. The captions are often copied directly from the source and lack sufficient explanation or interpretation from the authors, which limits their effectiveness in supporting the text.
Conclusion
The paper is suitable for publication after minor revisions addressing the above issues.
Author Response
The paper explores a highly dynamic and impactful research area: quasi-2D perovskite light-emitting diodes, with a particular emphasis on interface engineering—an essential factor in optimizing device performance. The manuscript is clearly structured and logically organized. Overall, the article is well-written and easy to follow, with a comprehensive overview of recent advances in the field.
However, there are a few aspects that require improvement:
- Several key concepts such as phase engineering, defect passivation, and additive strategies are discussed repeatedly across different sections without providing substantial new insights. The manuscript would benefit from a more concise and streamlined structure to reduce redundancy and enhance clarity.
Response: Thank you for bringing this important issue to our attention. We have noticed that these words are indeed repeatedly used in many places. Due to the fact that many of these problems/methods are interrelated, some methods may be able to solve two to three problems. We believe that these are excellent interface engineering methods and have made a lot of explanations, so there may be a problem of repeated application. We have also removed unnecessary duplicates and hope to receive your understanding.
- Many of the figures are reprinted from previous publications but are not well integrated into the main narrative. The captions are often copied directly from the source and lack sufficient explanation or interpretation from the authors, which limits their effectiveness in supporting the text.
Response: Thank you for this important issue. We acknowledge the concern regarding suboptimal figure clarity and have systematically revised all figures to ensure enhanced readability. We apologize for any confusion this may have caused. Thank you again for your valuable feedback. We hope this clarification addresses your concern.
We sincerely thank you for your valuable comments and suggestions, which have greatly improved the quality of our manuscript. We apologize for any oversights in the original submission and assure you that the review has been carefully revised for clarity and accuracy. On behalf of my co-authors, we express our deep gratitude for your contributions to our work.

Round 2
Reviewer 1 Report
Comments and Suggestions for Authors
Accepted
Reviewer 2 Report
Comments and Suggestions for Authors
This manuscript has been improved and may now be suitable for publication in Nanomaterials.